# An updated analysis of opioids increasing the risk of fractures

Qiaoning Yue[1], Yue Ma[2], Yirong Teng[3], Yun Zhu[4], Hao Liu[5], Shuanglan Xu[6], Jie Liu[6], Jianping Liu[7], Xiguang Zhang[1]*, Zhaowei Teng[1]*

1 Department of Orthopedic Surgery, The People's Hospital of Yuxi City, The 6th Affiliated Hospital of Kunming Medical University, Yuxi, Yunan, China, 2 Department of Pharmacy, The People's Hospital of Yuxi City, The 6th Affiliated Hospital of Kunming Medical University, Yuxi, Yunan, China, 3 Department of General Medicine, The People's Hospital of Yuxi City, The 6th Affiliated Hospital of Kunming Medical University, Yuxi, Yunan, China, 4 Department of Health Screening Center, The People's Hospital of Yuxi City, The 6th Affiliated Hospital of Kunming Medical University, Yuxi, Yunan, China, 5 Department of Emergency Medicine, The People's Hospital of Yuxi City, The 6th Affiliated Hospital of Kunming Medical University, Yuxi, Yunan, China, 6 Department of Respiratory Medicine, The Fourth Affiliated Hospital of Kunming Medical University, The Second People's Hospital of Yunnan Province, Kunming, China, 7 Department of Science and Education, The People's Hospital of Yuxi City, The 6th Affiliated Hospital of Kunming Medical University, Yuxi, Yunan, China

* tengzhaowei2003@163.com (ZT); gwkzxg@163.com (XZ)

## Abstract

### Objective

To assess the relationship between opioid therapy for chronic noncancer pain and fracture risk by a meta-analysis of cohort studies and case-control studies.

### Methods

The included cohort studies and case-control studies were identified by searching the PubMed and EMBASE databases from their inception until May 24, 2019. The outcome of interest was a fracture. This information was independently screened by two authors. When the heterogeneity among studies was significant, a random effects model was used to determine the overall combined risk estimate.

### Results

In total, 12 cohort studies and 6 case-control studies were included. We used the Newcastle-Ottawa Scale (NOS) to evaluate the quality of the included literature, and 14 of the studies were considered high-quality studies. The overall relative risk of opioid therapy and fractures was 1.78 (95% confidence interval (CI) 1.53–2.07). Subgroup analyses revealed sources of heterogeneity, sensitivity analysis was stable, and no publication bias was observed.

### Conclusions

The meta-analysis showed that the use of opioids significantly increased the risk of fracture.

**Data Availability Statement:** All relevant data are within the paper and its Supporting Information files.

**Funding:** The study was supported by grants from the National Natural Science Foundation of China

(No. 81660156 and 81860167), and Joint special fund of Applied Fundamental Research of Kunming Medical University granted by Science and Technology Office of Yunnan, grant no. 2018FE001 (-175):2018FE001(-174):2014FZ048; 2017FE468 (-181), Yunnan Special Funds for training high-level health and family planning technical personnel, grant no. H-2017064:H-2017028.

**Competing interests:** The authors have declared that no competing interests exist.

## Introduction

With the advancement of society, the number of elderly people has gradually increased. Pain is a common symptom in the elderly population, and the incidence of chronic pain ranges from 25% to 76% [1]. Opioids provide effective analgesic effects in a range of persistent noncancer pain conditions and are widely used for the treatment of noncancer pain due to their analgesic and psychoactive effects [2]. There are many side effects of using opioids, such as dizziness, hypogonadism, and inhibition of the innate and acquired immune system. These side effects can lead to fractures. Vestergaard et al. [3] revealed that opioid-induced fractures may be associated with vertigo in patients after opioid use. Grey et al. [4] also confirmed that opioids cause fractures and that opioids act on the gonads to reduce bone density [4].

In addition, studies have shown that opioid-induced fractures are associated with time of use [5] and are also associated with the use of opioids [6]. Opioids have been linked to the occurrence of fractures [2,3,7–11], and although the use of opioids has been reported to increase the risk of fractures, the trend of using opioids continues to increase [12]. However, in the previous studies, due to the influence of sample size, types of research, etc., there may have been inconsistencies, and we aimed to reconfirm the correlation between opioids and fracture risk while incorporating subsequently published studies.

## Materials and methods

### Search strategy and data sources

A search was conducted from the inception of the PubMed and EMBASE databases until May 20, 2019, to find relevant research that met the requirements. We also searched the bibliographies of relevant articles to identify additional studies. We used the following search terms: (i) fracture ? [Title/Abstract] OR "Fractures, Bone"[Mesh]; (ii) opioid ? [Title/Abstract] OR "Analgesics, Opioid"[Mesh].

### Study selection

Studies were considered eligible if they met all of the following criteria: (i) presented original data from the study; (ii) evaluated the association of opioid use with fracture incidence; (iii) had opioids as the exposure of interest; and (iv) provided hazard ratios and odd ratios (HRs and ORs) or the adjusted relative risks (RRs) and the corresponding 95% confidence intervals (CIs). If the data were duplicated or the population was studied in more than one study, we included the study with the largest sample size and the most comprehensive outcome evaluation.

### Data extraction

Two investigators (YQN, ZXG) independently evaluated the eligibility of the studies retrieved from the databases based on the predetermined selection criteria. In addition, a cross-refer ence search of eligible articles was conducted to identify studies not found in the computerized search. These two authors independently extracted the following data: the first author's name; year of publication, patient ages, sample size, study regions, years of follow-up, study design, HR, OR or RR and the 95% CIs, and statistical adjustments for confounding factors. Any disagreements were resolved either by discussion or in consultation with the co-corresponding author (TZW). The Newcastle-Ottawa Scale (NOS) was used to evaluate the quality of the research [13].

## Statistical analyses

Our primary objective was to evaluate the use of opioids and the increased risk of fractures. We calculated total RR and 95% CI from the adjusted RRs, ORs or HRs and 95% CIs reported in the studies. ORs and HRs were considered to correspond to RRs. Using the Cochran Q and $I^2$ statistic to assess statistical heterogeneity [14], we also calculated the P value of the q test representing heterogeneity; if the P value was less than 0.10, there was heterogeneity among the studies. The fixed effects model was applied when $I^2 <50\%$ [15], otherwise, the random effects model was applied [16]; to further explore the source of heterogeneity, we also examined the study design, the study area and subfamily analysis of fracture types (i.e., any fracture, non-spine fracture, hip fracture). Additionally, Begg's rank correlation test and Egger's linear regression test were conducted to assess the extent of potential bias [17]. Finally, we conducted a sensitivity analysis to assess the stability of the analytical results by excluding each study to explore the impact of individual studies on the overall outcome [18]. The data analyses were conducted using STATA statistical software version 12.0 (STATA Corp. LLC, College Station, TX, USA).

## Results

### Literature search and study characteristics

Using predefined search strategies and inclusion criteria, a total of 18 studies were included and 1,134 articles of unrelated literature were excluded (from a total of 600 articles from PubMed and 552 articles from EMBASE) after a detailed reading of the title, abstract, and full text, and the 18 articles included 884054 participants [3,7,9,19–32]. The detailed process of inclusion in this study is shown in Fig 1. Five studies were from the United States [12,21,23,24,26,32], three from Canada [9,19,28], two from the United Kingdom [18,27], one study was from Australia, and the remaining came from European countries; 12 articles were from cohort studies, and six were from case-control studies. The study information is shown in Table 1.

### Main analysis

There was a positive correlation between the use of opioids and fractures (RR 1.78, 95% CI 1.53–2.07) (Fig 2), and we observed significant heterogeneity among the studies. Eleven studies provided data on opioid use and hip fracture risk [3,7,19–23,27,29–31]. Pooled studies showed that the use of opioids had a significant impact on the risk of hip fracture (RR 1.56, 95% CI 1.37–1.79), and there was significant heterogeneity among the studies (P = 0.000, $I^2 = 83.1\%$) (Fig 3); we subsequently revealed the sources of heterogeneity through subgroup analyses.

### Subgroup meta-analysis

We performed a subgroup analysis based on the type of study, region, and fracture type, and the risk of fractures was positively correlated with the use of opioids (Table 2). Subgroup analyses showed a significant increase in fracture risk after opioid use, with no statistical heterogeneity among studies conducted in the European region, Britain, and Denmark (Fig 4). Although Shorr et al. [19] was a case-control study, the control data were derived from the hospital database, which make ita retrospective study together with the studies of Miller, Laura, Grewal, Kristine and Vakharia et al. [9,24,26,29,32]. To determine the impact of these retrospective studies, we conducted further analyses without the above studies, and the overall results showed that heterogeneity significantly decreased.

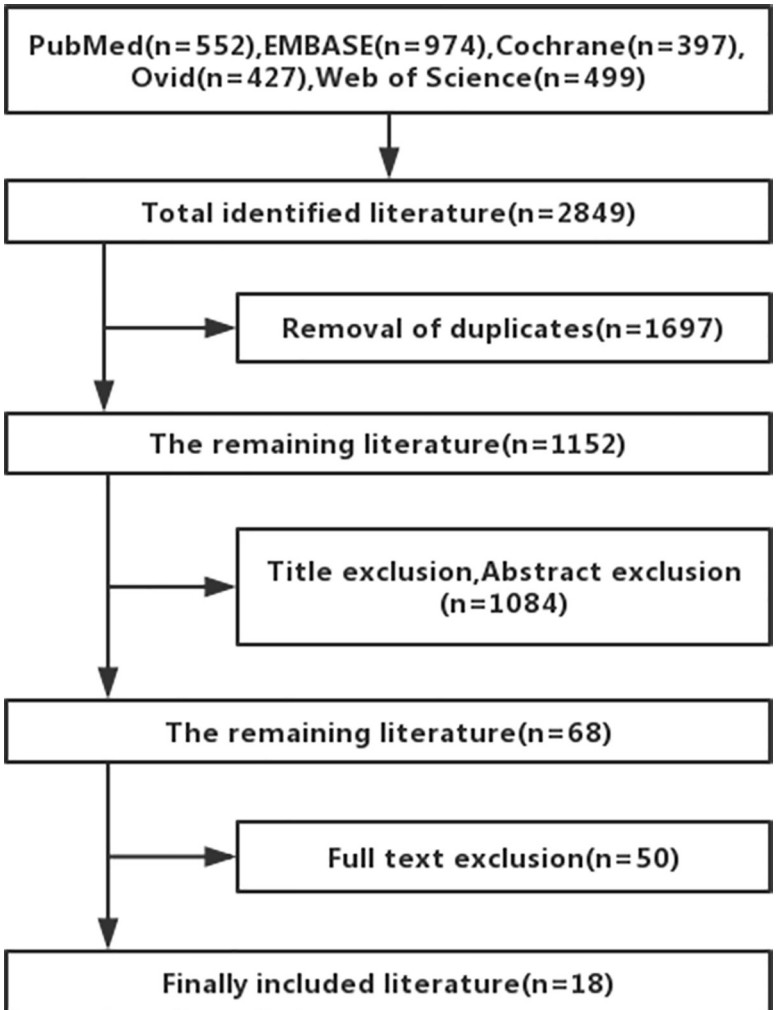

**Fig 1. Inclusion of literature search flow chart.**

## Sensitivity analysis

To assess the stability of our results, based on the original data, sensitivity analyses were performed using a strategy that systematically excluded individual studies. In the end, there was no change in the overall results (Fig 5).

## Publication bias

No evidence of publication bias was found based with Begg's rank correlation test (p>|z| = 0.649) or Egger's linear regression test (p>|z| = 0.067) (Figs 6 and 7).

## Discussion

The trend of population aging is becoming more pronounced, and most of the fracture patients are elderly individuals. The elderly population has a higher fracture rate due to lower bone density. Elderly individuals are more likely to be in poor physical condition, most of them have a history of chronic pain resulting in a history of taking opioids, and the probability

**Table 1. Basic characteristics of the 18 included studies.**

| Author, year, location | Age, years | Fracture type /assessment | Study design | Sample size | Follow-up time | Models | Adjustment for covariates | NOS |
|---|---|---|---|---|---|---|---|---|
| Jensen, 1991, Denmark | >59 | hip/WHO code 820 | Case-control | 400 | from April to December 1988 | Cornfield's iterative method | Age, sex, nursing home residency and number of hospital admissions | 6 |
| Shorr, 1992, Canada | ≥65 | hip/ ICD-8, ICD-9 | Case-control | 28541 | from 1997 to 1985 | Unconditional logistic regression | Age, sex, home, hospital discharge in preceding year, index year | 7 |
| Guo, 1998, Sweden | ≥75 | hip/ICD-9 | Prospective cohort | 1608 | 4.4 years | Cox proportional hazards | Age, sex, education, residence, ADL limitation, cognitive impairment, history of stroke and tumors | 8 |
| Ensrud, 2003, USA | ≥65 | fractures/ radiology reports | Prospective cohort | 8127 | 4.8 years | Cox proportional hazards | Age, sex, race, health status, smoking, walking exercise, functional impairment, cognitive function, depression, weight change | 9 |
| Card, 2004, UK | NA | hip/NA | Prospective cohort | 99467 | 7.3 per 10000 person-years | Cox regression | Age, sex, practice, corticosteroid use | 6 |
| Sachin, 2006, USA | ≥65 | hip/ICD-9 | Prospective cohort | 362503 | 464 days | Cox regression | Age, sex, use of antidepressants, antipsychotics, anxiolytics/hypnotics | 7 |
| Vestergaard, 2006, Denmark | 43.44 ± 27.39 | hip/NA | Case-control | 42065 | during 2006 | Conditional logistic regression | Use of other drugs | 6 |
| Kathleen, 2010, USA | ≥60 | fractures/ ICD-9 | Prospective cohort | 2341 | 32.7 months | Cox proportional hazards | Age, gender, smoking, depression, substance abuse, dementia, comorbidity, prior fracture, pain site, antidepressant use, sedative use, HRT/ bisphosphonate use | 9 |
| Miller, 2011, USA | ≥65 | fractures/ ICD-9 | Retrospective cohort | 17310 | 451 per 1000 person-years | Cox proportional hazards | Age, sex, diabetes, stroke, osteoarthritis, comorbidity index, stroke, diabetes | 6 |
| Vestergaard, 2012, Denmark | 45 to 58 | Fractures /X-ray | Prospective cohort | 2016 | 10 years | Cox proportional hazards | Age, HT, BMI, baseline spine bone mineral density (BMD), family or prior fracture, serum 25-hydroxy-vitamin levels and smoking | 9 |
| Laura, 2013, USA | ≥58.73 ±13.43 | lower extremity /ICD-9 | Retrospective cohort | 7447 | 3–8 years | Cox proportional hazards | Age, race, completeness of spinal cord injury (SCI) level and duration of SCI | 7 |
| Lin Li, 2013, UK | 18 to 80 | fracture/NA | Nested case-control | 71538 | from 1990 to 2008 | Conditional logistic regression | Smoking, BMI, comorbidities. Number of general practice visits recorded during the years before index date | 7 |
| Kristine, 2014, Sweden | ≥75 | hip/codes S72.0, S72.1, S72.2 | Retrospective cohort | 38407 | during 2006 | Multivariate logistic regression | Age, gender and morbidity level | 8 |
| Leach, 2015, Australia | >65 | hip/ICD codes S72.0 or S72.1 | Case-crossover | 8828 | from 2009 to 2012 | Conditional logistic regression | NA | 8 |
| Acurcio, 2016, Canada | 76.33 ±10.04 | fracture/ICD-9, ICD-10 | Retrospective nested case-control | 9769 | from 2007 to 2012 | Conditional logistic regression | Age, sex, measures of comorbidities, history of arthroplasty, corticosteroid use, biologic agents or traditional disease-modifying antirheumatic drugs (DMARDs), use of other drugs potentially influencing the risk of fractures or falls, measures of health care resource use | 7 |
| Grewal, 2018, Canada | ≥65 | fracture/ICD-10 | Retrospective cohort | 89897 | 3 months | Cox regression | Age, sex, past medical history, health care use, etc. | 7 |
| Taipale, 2018, Finland | NA | hip fracture/ ICD-10 | Retrospective matched cohort | 70718 | 5 years | Cox proportional hazard | Age, sex, time since Alzheimer's disease (AD) diagnosis, socioeconomic position, university hospital catchment area, use of drugs, comorbidities | 9 |
| Vakharia, 2019, USA | ≥64 | fracture/ICD-9 (81.54) codes 304.00–304.02 and 305.50–305.52. | Retrospective matched cohort | 23072 | from 2005 to 2014 | R Statistical analysis | Age, sex, use of drugs | 7 |

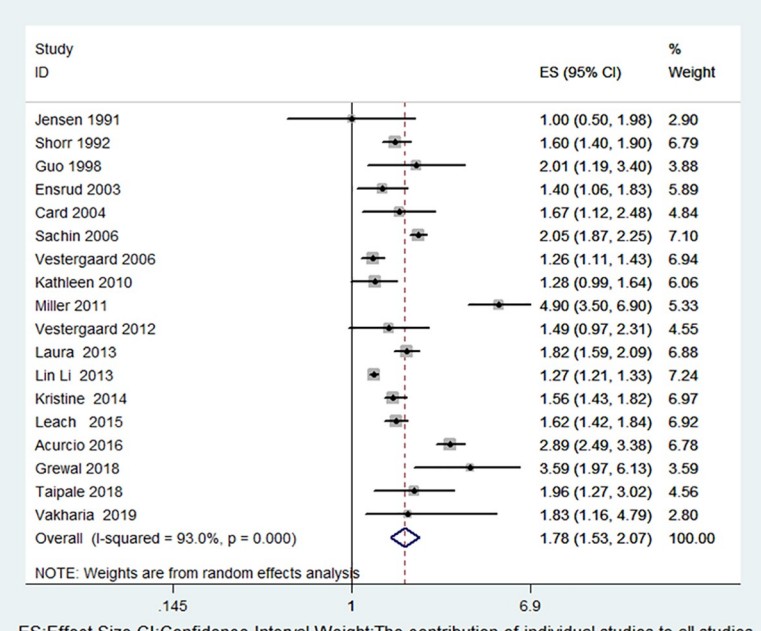

**Fig 2. Forest plot of RR with 95% CI for opioid use and fracture risk.**

of fractures increases. Therefore, the incidence of fractures caused by opioids is discussed below. There is a high potential for associations between opioids and fractures.

In this meta-analysis, we included the latest basic research. The types of studies included in this analysis included case-control studies, the sample size was increased, and the study area was refined. The results showed that the use of opioids increased the risk of fracture. Previously, the most recent meta-analysis (Ping et al. [33]) was limited to the study of hip fractures,

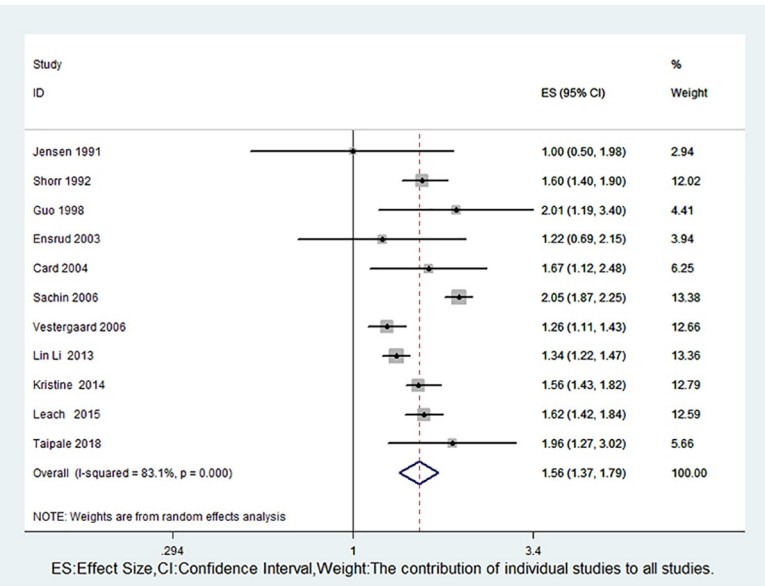

**Fig 3. Forest plot of RR with 95% CI for opioid use and hip fracture risk.**

**Table 2. Subgroup analyses of the association between opioid use and fracture risk.**

|  | Factor | No. of studies | RR (95% CI) | Heterogeneity P (I2%) |
|---|---|---|---|---|
| Study design | Case-control | 6 | 1.57 (1.23, 2.02) | 0.000 (95.6) |
|  | Prospective cohort | 6 | 1.62 (1.31, 2.02) | 0.003 (72.5) |
|  | Retrospective cohort | 6 | 2.32 (1.69, 3.19) | 0.000 (93.0) |
| Fracture type | Hip fracture | 9 | 1.62 (1.41, 1.87) | 0.000 (81.3) |
|  | Nonspine fracture | 2 | 2.03 (1.00, 4.13) | 0.000 (95.1) |
|  | Any fracture | 7 | 1.97 (1.43, 2.69) | 0.000 (93.5) |

and Grewal et al. [9] showed that patients taking opioids had a risk of fracture after discharge compared with patients who were not taking opioids. The main reason for the increase was that patients taking opioids were prone to vertigo and falls that can lead to fractures. In addition, Aspinall's et al. [6] study showed that patients receiving opioid therapy had an increased risk of falls, and the accompanying final outcomes were fractures [6]. Schwarzer et al's [8] study also suggested that when opioid use was considered, the risk of fractures increased [8]. The above studies are consistent with our final results and support our findings. In addition, there are two main mechanisms for the occurrence of fractures with opioid use. One mechanism is that opioids may reduce bone density by inhibiting the production of endogenous sex hormones, leading to an increased risk of fractures [34]; the other mechanism involves the side effects of opioids, such as the central nervous system side effects of vertigo, fatigue, etc., that lead to the occurrence of fractures [7,9,27,28], and there is a high incidence of side effects,

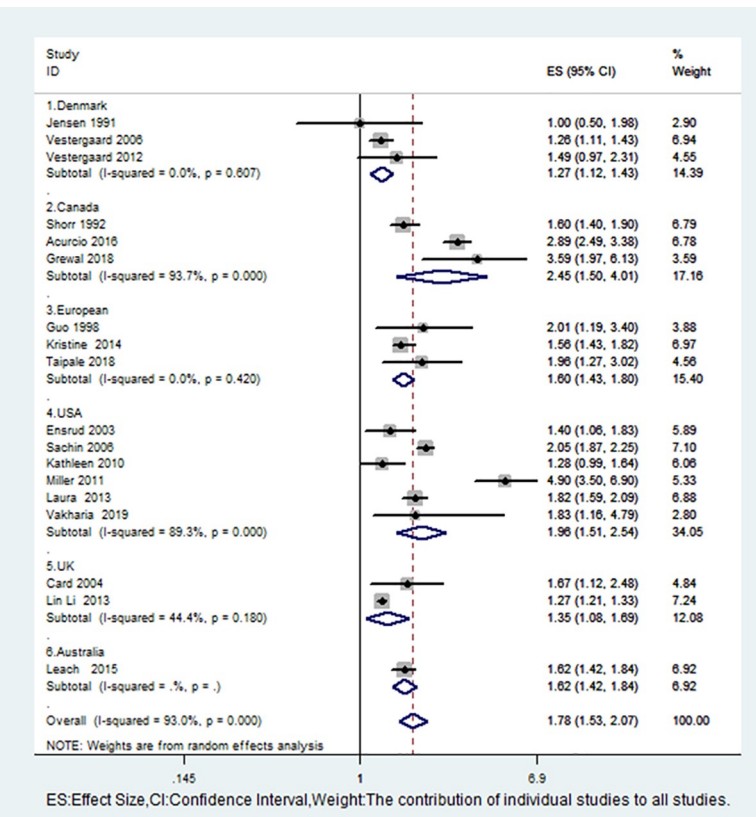

**Fig 4. Forest plot for a subgroup meta-analysis by region.**

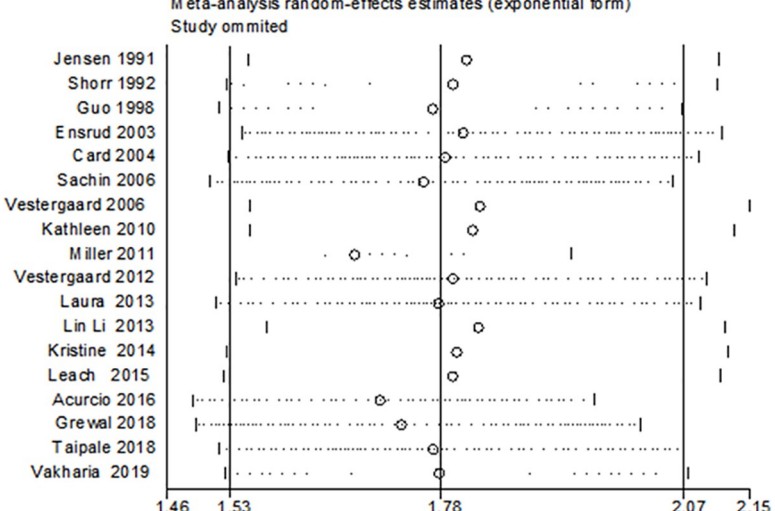

**Fig 5. Sensitivity analysis of the association between opioid use and fracture risk.**

including acute cognitive deterioration, increased sputum production, decreased oxygen saturation, and constipation, after the use of opioids in elderly populations, as confirmed in recent studies [35]. The trend of the population aging is becoming increasingly pronounced, and osteoporosis in this aging population is a serious concern. The use of opioids in this population leads to more frequent fractures.

The relationships among opioids, analgesia and fracture have been examined, and we have previously published relevant articles [36]. However, for the present analysis, we included cohort studies and case-control studies, in which patients were followed up over a long time. Most of the research was of high quality, the sample sizes were large enough, and the outcome evaluations were reliable and comprehensive. In addition, although our overall analysis showed heterogeneity, we determined the source of the heterogeneity through subgroup analyses. For example, Shorr, Grewal, Miller, and Laura were all retrospective studies [9,19,24,26].

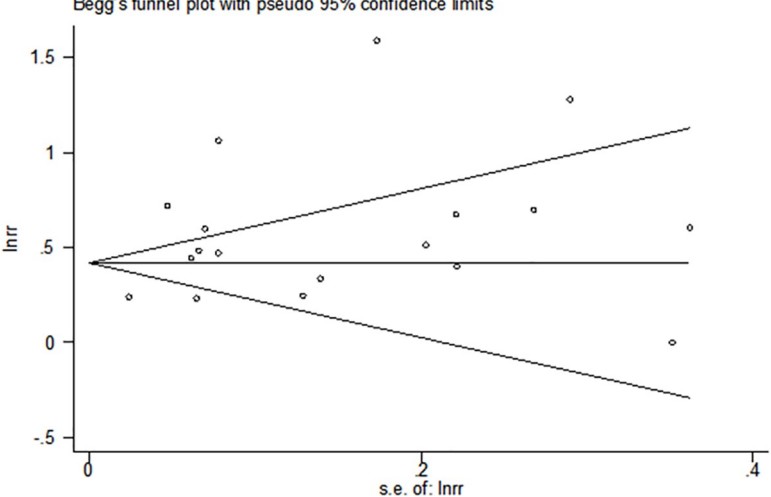

**Fig 6. Begg's funnel plot.**

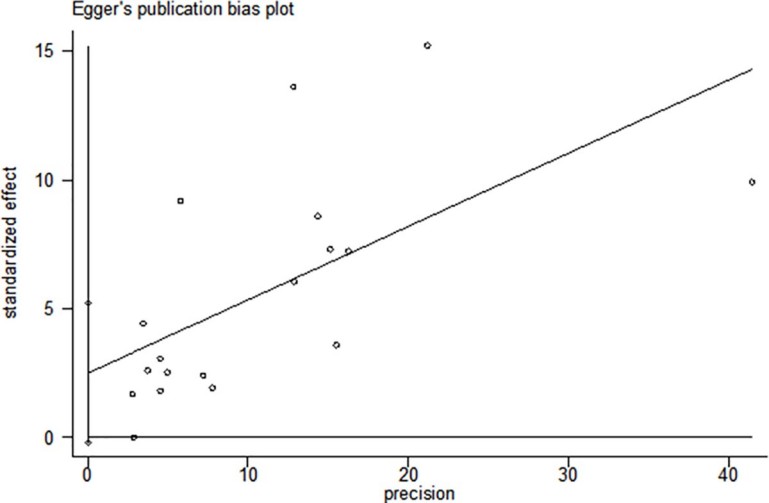

**Fig 7. Egger's publication bias plot.**

In the subgroup analysis, heterogeneity was significantly reduced suggesting that these retrospective cohort studies may have been a source of heterogeneity. Based on a regional subgroup analysis, we found that the research conducted in Canada and the United States made an important contribution to the heterogeneity (Fig 5). Therefore, we believe that geography is one of the important reasons for the heterogeneity. Next, we individually examined the heterogeneity in the Canadian group of studies. When we excluded the study by Shorr et al. [19], we found that there was no heterogeneity among the Canadian group of studies (I2 = 0.0%, p = 0.470) and in the hip fracture group of studies. Regarding the larger source of heterogeneity, we finally found the source through analysis and mainly identified the role of retrospective cohort studies and case-control studies. This comprehensive analysis suggested that heterogeneity mainly comes from retrospective articles and may also be caused by other factors, such as geographical factors, and we will continue to pay attention to these factors in the future.

Although our research has many advantages, it also has shortcomings. First, due to language limitations, the included studies were limited to English, and these language limitations may have led to studies not being included, resulting in a dataset that was not quite comprehensive. Second, some studies that are not statistically significant or have negative findings may not have been published because they were rejected by the journal or because the researcher was unwilling to submit such a publication. We also performed a publication bias test, but it is also possible that the effect value was overestimated when studies with a large degree of heterogeneity were combined. Again, the degree of control over confounding variables, such as age and gender, varied from study to study. In our meta-analysis, the timing and dose of the drug could not be studied because the time frames were different across studies. Thus, we were unable to unify the timing, and the drug dose was also different based on varying classification criteria and could not be further studied. Finally, the study participants were all Westerners, and the influencing factors were complex and variable. Therefore, we should pay attention to the global situation in these populations to improve and validatethe conclusions. It was also impossible to conduct further analyses as to whether the length of metabolism for a particular drug was an influencing factor, and this issue is worthy of attention in the future. We have included a number of different studies covering a wide range clinical and experimental factors and the results were still stable, and we will conduct a more comprehensive analysis when future conditions permit.

## Conclusions

Taken together, we included different types of studies, and the results still indicated that the use of opioids significantly increased the risk of fracture. Further research, including well-designed international trials, studies of the mechanisms by which opioid use causes fractures, and studies aimed at preventing such fractures require more evidence from clinical practice.

## Supporting information

**S1 Table. PRISMA 2009 checklist.**
(DOC)

**S1 File. Editorial certificate.**
(PDF)

**S1 Data.**
(DOC)

## Acknowledgments

We appreciate the contribution of all patients, their families, the investigators and the medical staff.

## Author Contributions

**Conceptualization:** Hao Liu, Jianping Liu.

**Data curation:** Qiaoning Yue, Yue Ma, Yirong Teng, Hao Liu, Zhaowei Teng.

**Formal analysis:** Qiaoning Yue, Yirong Teng, Yun Zhu, Jianping Liu, Xiguang Zhang.

**Funding acquisition:** Yun Zhu, Xiguang Zhang, Zhaowei Teng.

**Investigation:** Yun Zhu, Shuanglan Xu, Jianping Liu.

**Methodology:** Qiaoning Yue, Yun Zhu, Shuanglan Xu, Xiguang Zhang, Zhaowei Teng.

**Project administration:** Xiguang Zhang.

**Resources:** Yue Ma, Yirong Teng, Shuanglan Xu.

**Software:** Qiaoning Yue, Shuanglan Xu, Jie Liu, Zhaowei Teng.

**Supervision:** Yue Ma, Hao Liu, Jie Liu.

**Validation:** Yue Ma, Jie Liu.

**Visualization:** Yue Ma, Hao Liu.

**Writing – original draft:** Qiaoning Yue, Xiguang Zhang, Zhaowei Teng.

**Writing – review & editing:** Qiaoning Yue, Yue Ma.

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
