## [Decision Letter · Decision Letter 0]

26 Sep 2019

PONE-D-19-19172

An updated analysis of opioids increasing the risk of fractures

PLOS ONE

Dear Dr Teng,

Thank you for submitting your manuscript to PLOS ONE. After careful consideration, we feel that it has merit but does not fully meet PLOS ONE’s publication criteria as it currently stands. Therefore, we invite you to submit a revised version of the manuscript that addresses the points raised during the review process.

We would appreciate receiving your revised manuscript by Oct 18 2019 11:59PM. To enhance the reproducibility of your results, we recommend that if applicable you deposit your laboratory protocols in protocols.io, where a protocol can be assigned its own identifier (DOI) such that it can be cited independently in the future. For instructions see: http://journals.plos.org/plosone/s/submission-guidelines#loc-laboratory-protocols

We look forward to receiving your revised manuscript.

Kind regards,

Maw Pin Tan, M.D.

Academic Editor

PLOS ONE

Journal Requirements:

2. We note that you have reported significance probabilities of 0 in places. Since p=0 is not strictly possible, please correct this to a more appropriate limit, eg 'p<0.0001'.

3. We noticed you have some minor occurrence(s) of overlapping text with the following previous publication(s), which needs to be addressed:

https://doi.org/10.1371/journal.pone.0128232

In your revision ensure you cite all your sources (including your own works), and quote or rephrase any duplicated text outside the Methods section. Further consideration is dependent on these concerns being addressed.

Reviewers' comments:

Reviewer's Responses to Questions

**Comments to the Author**

1. Is the manuscript technically sound, and do the data support the conclusions?

Reviewer #1: Partly

Reviewer #2: Yes

Reviewer #3: No

2. Has the statistical analysis been performed appropriately and rigorously? 

Reviewer #1: Yes

Reviewer #2: I Don't Know

Reviewer #3: No

3. Have the authors made all data underlying the findings in their manuscript fully available?

Reviewer #1: Yes

Reviewer #2: Yes

Reviewer #3: Yes

4. Is the manuscript presented in an intelligible fashion and written in standard English?

Reviewer #1: Yes

Reviewer #2: Yes

Reviewer #3: No

5. Review Comments to the Author

Reviewer #1: Yue et al. intended to perform meta-analysis to assess the relationship between opioid therapy for chronic non-cancer pain and fracture risk. They analyzed 18 studies including 14 classified as high-quality studies. The results showed that the use of opioid therapy and fractures was significantly associated.

1. Line 64. Authors mentioned that there have been inconsistencies association results between risk of fractures and opioid use. However, only positive result was cited with reference. Negative results should be cited to support their argument. if no negative results were reported, their motivation will be invalid and leaves the contribution of this work questionable.

2. Please define fracture in the text as well. i.e. what type of fractures were considered in this study? Different fracture may have different mechanisms.

3. What’s NOS in Table 1? The abbreviation needs to be spelled out somewhere with explanation.

4. Table 1. Need to provide the reference citation for each paper as well.

5. Figures. Footnotes are needed. e.g. what’s ES, Weight and so on?

6. As there are so many factors different from one study to another, driving high heterogeneity, it’s questionable to perform the analysis across all studies as primary analysis.

Reviewer #2: All the active drugs on the central nervous system (CNS) administered to osteoporotic elderly patients can determine fall-related injuries. Often falls have many different causes resulting from the interactions between intrinsic or extrinsic risk factors. The intrinsic risks as functional impairment or balance disorders represent the common features of the frail elderly osteoporotic patient. The extrinsic risks are often linked to treatment as the adverse drug reaction. Several studies have documented that there is a relation between falling and the number of drugs used. Drugs with central nervous system (CNS) side effects, such as benzodiazepines, antidepressants, neuroleptics, anticonvulsants and opioids are known to increase the risk of fractures and fall-related injuries. Frequently the CNS effects of opioids happen

starting opioid therapy or during substantial dose escalation.

Almost always after a few days of treatment, opioids tolerance relives CNS symptoms. Although opioids can be

essential in the treatment of moderately severe chronic pain.

Reviewer #3: Introduction

Line 54-55: It is worth including “gut dysfunction” as one of the side effects of opioid use.

Line 55-56: The authors sate that “These side effects can lead to fractures”. This sentence is spurious. Should briefly elaborate how the side effects could increase fractures or be associated with fractures.

The rationale for the study in the introduction is poorly described and there is a huge lack of logical flow in this section. The introduction should be re-written nailing the urge for an update, with valid references.

Methods & Results

I wonder why authors of this manuscript has only limited the search for two databases given the prevalence of evidence in this field.

I’m afraid that the search terms used will not assist in capturing all research in this context. I suggest to re-run the search using the word “pain” as well.

The selection criteria for the studies are poorly elaborated and the reader is not helped on how the studies have been selected for the analysis. The flow of logical consistency is lost while progressing from one paragraph to the other.

I would strongly encourage the authors to re-write the methods section in accordance to current scientific standards

The PRISMA flow diagram should be revised. The terminology used in each stage of the process is not current. For instance, “removal of literature from reading headings” should be termed as” Title exclusion”; “removal of repeat literature” should be termed as “removal of duplicates”

Authors have used Table 1 to describe the study characteristics, however, the representation of this table is not up to the expected standard. Needs revision. The font and size of the text in the table is not comparable to the text in the manuscript.

Discussion

The discussion doesn’t read well, and the flow doesn’t seem logical and not connected to the method section very well. Therefore, I recommend redoing the discussion section as well.

General points,

• The methodological reporting mars the study quality

• The reference cited is not support of the findings reported

• Authors really do need to take care of the technical words used in the manuscript.

• There are grammar mistakes and typos throughout the manuscript, and I would encourage the authors to get support from a native English speaker to improve the clarity of the language.

6. PLOS authors have the option to publish the peer review history of their article (what does this mean?). If published, this will include your full peer review and any attached files.

Reviewer #1: No

Reviewer #2: Yes: Renato Vellucci

Reviewer #3: No

---

## [Author Response · Author response to Decision Letter 0]

4 Nov 2019

Dear Editor,

We are truly grateful to you and the other reviewers for the critical comments and thoughtful suggestions. We have made careful modifications to the original manuscript based on these comments and suggestions, and all changes made to the manuscript are in red,and the modified and unmodified files are named 'Revised Manuscript with Track Changes'、 'Manuscript' upload. Our financial disclosure has not changed.. We hope the revised manuscript will meet your journal’s standard. Below you will find our point-by-point responses to the reviewers' comments:

Journal Requirements: 

1.Please ensure that your manuscript meets PLOS ONE's style requirements, including those for file naming.

Response: Thank you for your comment. We has been read the relevant requirements of your magazine again, and adjusted the uploaded file name, image format, manuscript requirements, etc. one by one to meet the relevant requirements of your magazine. I hope that our adjustments will reduce unnecessary trouble for editors and reviewers.

2.We note that you have reported significance probabilities of 0 in places. Since p=0 is not strictly possible, please correct this to a more appropriate limit, eg 'p<0.0001'.

Response: Thank you for your comment. We have changed the 'p=0.000' to a more appropriate limit, eg 'p<0.0001'.

3.We noticed you have some minor occurrence(s) of overlapping text with the following previous publication(s), which needs to be addressed:

https://doi.org/10.1371/journal.pone.0128232

Response: Thank you for your comment. In our new manuscript, we have solved the problem of a slight overlap with the previous article.

Reviewers' comments:

1. Is the manuscript technically sound, and do the data support the conclusions?

Response:Thank you for your comment.We conduct our research in strict accordance with the standards of current scientific research. And deal with statistical data objectively.

2.Has the statistical analysis been performed appropriately and rigorously?

Response:Thank you for your comment.In the data processing, we seek truth from facts and strictly follow the objective principles of statistical analysis. If there is a flaw, we sincerely hope to receive your opinion again.

3.Have the authors made all data underlying the findings in their manuscript fully available?

Response:Thank you for your comment.The authors have completely disclosed all the data in their manuscripts.

4.Is the manuscript presented in an intelligible fashion and written in standard English?

Response:Thank you for your comment.When we finished the manuscript, we got help from a native English speaker.

5.Review Comments to the Author

Response:Thank you for your comment.The reviewers gave us the best advice, and we did find out the shortcomings of our existence from the opinions or suggestions of the reviewers. We have also briefly explained the above issues, and sincerely thank all the reviewers.

Reviewer #1: 

1.Line 64. Authors mentioned that there have been inconsistencies association results between risk of fractures and opioid use. However, only positive result was cited with reference. Negative results should be cited to support their argument. if no negative results were reported, their motivation will be invalid and leaves the contribution of this work questionable.

Response:Thank you for your comment.In the latest manuscript, we have cited references to negative results.

2.Please define fracture in the text as well. i.e. what type of fractures were considered in this study? Different fracture may have different mechanisms.

Response:Thank you for your comment.In the latest manuscript, we redefine the type of fracture. The specific content we have stated in the introduction.

3.What’s NOS in Table 1? The abbreviation needs to be spelled out somewhere with explanation.

Response:Thank you for your comment.The abbreviation of NOS in Table 1 refers to Newcastle-Ottawa Scale. We have already explained below in Table 1.

4.Table 1. Need to provide the reference citation for each paper as well.

Response:Thank you for your comment.In our latest manuscript, we have provided a reference citation for each article in Table 1.

5.Figures. Footnotes are needed. e.g. what’s ES, Weight and so on?

Response:Thank you for your comment.The numbers and footnotes in the picture have been provided to the latest attachments.

6.As there are so many factors different from one study to another, driving high heterogeneity, it’s questionable to perform the analysis across all studies as primary analysis.

Response:Thank you for your comment.First, two types of research we have included are observational studies.Second, in our research events, the incidence is small,Most of studies were of high quality.Finally, there are certain deficiencies in our approach, such as research bias, confounding effects, etc., but we focus on the source of heterogeneity and ultimately determine that the type of research is not the main source of heterogeneity. For this issue, we will carry out deeper research. Thank you again for your advice.

Reviewer #2: 

Response:Sincerely thank you for your comments.In the future, we will continue to pay attention to the risk relationship between central nervous system drugs and other drugs and fall-related fractures. And further concerned about the impact of the starting dose of the drug, the time of the drug.

Reviewer #3: 

Introduction

1.Line 54-55: It is worth including “gut dysfunction” as one of the side effects of opioid use.

Response:Sincerely thank you for your comments.We have listed “gut dysfunction” as one of the side effects of opioids as required.

2.Line 55-56: The authors sate that “These side effects can lead to fractures”. This sentence is spurious. Should briefly elaborate how the side effects could increase fractures or be associated with fractures.

Response:Sincerely thank you for your comments.In the latest manuscript, we have briefly described the relationship between opioid side effects and fractures.

3.The rationale for the study in the introduction is poorly described and there is a huge lack of logical flow in this section. The introduction should be re-written nailing the urge for an update, with valid references.

Response:Sincerely thank you for your comments.After carefully reading the introduction, we added valid references and re-written them. Thank you again for your advice.

Methods & Results

1.I wonder why authors of this manuscript has only limited the search for two databases given the prevalence of evidence in this field.

Response:Sincerely thank you for your comments.At the beginning, the authors of the manuscript believed that the literature in the two databases was comprehensive, but we used the reviewers' comments to re-search the other three databases (Cochrane; Ovid; Web of Science).

2.I’m afraid that the search terms used will not assist in capturing all research in this context. I suggest to re-run the search using the word “pain” as well.

Response:Thank you for your comment.After listening to your suggestion, we added the word "pain" to PubMed; EMBASE; Cochrane; Ovid; Web of Science's five major databases for re-search, but the number of studies we eventually included did not change.

3.The selection criteria for the studies are poorly elaborated and the reader is not helped on how the studies have been selected for the analysis. The flow of logical consistency is lost while progressing from one paragraph to the other.

Response:Thank you for your comment.We have redefined the inclusion criteria for the study. The details are presented in our latest manuscript.

4.I would strongly encourage the authors to re-write the methods section in accordance to current scientific standards

Response:Thank you for your comment.We have rewritten the method section and hope that you can give valuable feedback again.

5.The PRISMA flow diagram should be revised，. The terminology used in each stage of the process is not current. For instance, “removal of literature from reading headings” should be termed as” Title exclusion”; “removal of repeat literature” should be termed as “removal of duplicates”.

Response:Sincerely thank you for your comments.We have changed the "removal of literature from reading headings" to "title exclusion"; "removal of repeat literature" to "removal of duplicates" in PRISMA.

6.Authors have used Table 1 to describe the study characteristics, however, the representation of this table is not up to the expected standard. Needs revision. The font and size of the text in the table is not comparable to the text in the manuscript.

Response:Sincerely thank you for your comments.We have re-edited the format and fonts of Table 1 as required. The revised Table 1 is in our latest manuscript.

Discussion

The discussion doesn’t read well, and the flow doesn’t seem logical and not connected to the method section very well. Therefore, I recommend redoing the discussion section as well.

Response:Sincerely thank you for your comments.After discussing with other authors, we rewrote the discussion. Make sure that the discussion and method parts are more closely linked and more logical. Thanks again for your advice.

General points

• The methodological reporting mars the study quality；

• The reference cited is not support of the findings reported；

• Authors really do need to take care of the technical words used in the manuscript.；

• There are grammar mistakes and typos throughout the manuscript, and I would encourage the authors to get support from a native English speaker to improve the clarity of the language.

Response:Sincerely thank you for your comments.After discussing with all the authors, we adopted the suggestions of the reviewers and carefully and carefully revised the articles. The grammatical errors in the manuscripts were also corrected. I hope that our modifications are effective. We sincerely look forward to your reply.

We look forward to receiving your reply.

Yours sincerely,

Zhaowei Teng

---

## [Decision Letter · Decision Letter 1]

9 Mar 2020

An updated analysis of the increased risk of fractures with opioids

PONE-D-19-19172R1

Dear Dr. Teng,

We are pleased to inform you that your manuscript has been judged scientifically suitable for publication and will be formally accepted for publication once it complies with all outstanding technical requirements.

With kind regards,

Maw Pin Tan, M.D.

Academic Editor

PLOS ONE

Additional Editor Comments (optional):

Reviewers' comments:

Reviewer's Responses to Questions

**Comments to the Author**

1. If the authors have adequately addressed your comments raised in a previous round of review and you feel that this manuscript is now acceptable for publication, you may indicate that here to bypass the “Comments to the Author” section, enter your conflict of interest statement in the “Confidential to Editor” section, and submit your "Accept" recommendation.

Reviewer #1: All comments have been addressed

2. Is the manuscript technically sound, and do the data support the conclusions?

Reviewer #1: (No Response)

3. Has the statistical analysis been performed appropriately and rigorously? 

Reviewer #1: (No Response)

4. Have the authors made all data underlying the findings in their manuscript fully available?

Reviewer #1: (No Response)

5. Is the manuscript presented in an intelligible fashion and written in standard English?

Reviewer #1: (No Response)

6. Review Comments to the Author

Reviewer #1: (No Response)

7. PLOS authors have the option to publish the peer review history of their article (what does this mean?). If published, this will include your full peer review and any attached files.

Reviewer #1: No

---

## [Editor Report · Acceptance letter]

27 Mar 2020

PONE-D-19-19172R1 

An updated analysis of the increased risk of fractures with opioids 

Dear Dr. Teng:

I am pleased to inform you that your manuscript has been deemed suitable for publication in PLOS ONE. Congratulations! Your manuscript is now with our production department. 

With kind regards,

on behalf of

Dr. Maw Pin Tan 

Academic Editor

PLOS ONE